# InGaN Laser Diodes with Etched Facets for Photonic Integrated Circuit Applications

**DOI:** 10.3390/mi14020408

**Published:** 2023-02-09

**Authors:** Krzysztof Gibasiewicz, Anna Kafar, Dario Schiavon, Kiran Saba, Łucja Marona, Eliana Kamińska, Piotr Perlin

**Affiliations:** 1Institute of High Pressure Physics, Polish Academy of Sciences, Sokolowska 29/37, 01-142 Warsaw, Poland; 2TopGaN Limited, Sokolowska 29/37, 01-142 Warsaw, Poland

**Keywords:** InGaN laser, Fabry–Perot, wet etching GaN, GaN TMAH

## Abstract

The main objective of this work is to demonstrate and validate the feasibility of fabricating (Al, In) GaN laser diodes with etched facets. The facets are fabricated using a two-step dry and wet etching process: inductively coupled plasma—reactive ion etching in chlorine, followed by wet etching in tetramethylammonium hydroxide (TMAH). For the dry etching stage, an optimized procedure was used. For the wet etching step, the TMAH temperature was set to a constant value of 80 °C, and the only variable parameter was time. The time was divided into individual steps, each of 20 min. To validate the results, electro-optical parameters were measured after each step and compared with a cleaved reference, as well as with scanning electron microscope imaging of the front surface. It was determined that the optimal wet etching time was 40 min. For this time, the laser tested achieved a fully comparable threshold current (within 10%) with the cleaved reference. The described technology is an important step for the future manufacturing of photonic integrated circuits with laser diodes integrated on a chip and for ultra-short-cavity lasers.

## 1. Introduction

The interest in developing the technology of fabricating laser diode Fabry–Perot resonators by dry or wet etching is almost as old as the modern semiconductor laser diodes themselves [1,2,3,4]. Although the laser bars’ cleavage technology provides parallel and smooth mirrors [5,6], it is at the same time expensive, difficult to perform, and non-compatible with on-wafer testing procedures. The main motivation to form the mirrors by etching was and still is the fabrication of short-cavity lasers (short lasers are difficult to cleave) and photonic integrated circuits with fully integrated light sources. There are two fundamental problems related to etched laser mirror technology, namely, the verticality of the mirrors and their smoothness. Verticality requires the etch process to be sensitive to the crystallographic orientation of the facet, and smoothness may be problematic in the case of the etching multilayer structure (varying chemical composition) of the laser diodes.

In the early work of Iga et al. [2] and Miller et al. [3], the possibility of the formation of laser diodes mirrors by wet chemical etching using GalnAsP/lnP lasers was demonstrated. The etched lasers had their threshold currents higher by 50–100% as compared to standard cleaved counterparts. In the next decade, many researchers attempted to use various etching schemes for the GaAs family of lasers. The methods included focused ion beam etching (FIB) [7], reactive ion beam etching (RIBE) [8], chemical-assisted ion beam etching (CAIBE)’s [5,9], and reactive ion etching (RIE) [1,3,4,7,8,9,10,11,12,13]. The gradual improvement of lasing parameters was observed, concluded by the fabrication of high-power laser diodes with etched mirrors by the RIE method [14].

The advent of nitride semiconductors (AlInGaN) revived the interest in Fabry–Perot cavity formation by etching instead of cleavage. There are a couple of reasons for this: (i) problems with cleavage for structures grown on sapphire [15,16] or on silicon [17], (ii) interest in short-cavity lasers [16], (iii) poor cleavage of nitride laser diodes fabricated on semi or non-polar GaN substrate [16], and (iv) growing interest in photonic integrated circuits [18,19].

The etching of nitride semiconductors is usually achieved using RIE. Typically, the mixture of Ar and Cl is used to etch Ga containing semiconductors, as GaCl_2_ is easy to remove by the physical action of Ar ions. Of course, RIE or other dry etching processes are not the only possible way of structuring the semiconductor. However, it should be emphasized that for the fabrication of a laser facet, even taking into account further improvement of the facet verticality by wet etching, very high precision is required.

The typical depth of the etching is of the order of the thickness of the epitaxy, which is in our case of the order of 2 μm [20]. However, the roughness of the etching should also be clearly below the wavelength of emission of our laser (430 nm). One example of a possible method is focused ion beam (FIB). It can be used to prepare the laser facet with high precision, but this method requires “sculpting” of the material by the beam for every facet separately, while RIE is a method that modifies the whole crystal at the same time, which is faster and more suitable from a technological point of view. Another interesting method is atomic layer etching (ALE) [21], which is based on using a sequence of self-limiting chemical reaction steps that allow one to remove the material by a single atomic layer. Such a procedure can be treated as similar to the RIE procedure in the sense of uniformly treating the whole wafer. ALE is extremely precise, but at the same time it is significantly slower than RIE. However, achieving perfectly vertical facets is extremely challenging with this approach. A popular solution is to add an additional etching step of wet etching that is crystallographically selective [16,17,22]. This allows one to reveal the chosen plane, usually the m-plane.

In our work, we follow the approach of using tetramethylammonium hydroxide (TMAH) solution [16,17]. The mechanism of GaN etching by TMAH solution is explained by the attachment of hydroxide ions to the positively charged Ga dangling bonds, while adjacent negatively charged N dangling bonds will prohibit the hydroxide ions from accessing Ga atoms [23].

Within this work, our focus lies in optimizing the processing schemes for etched facet lasers and studying their work parameters. The lasers from the same wafer, obtained by traditional cleaving and lasers with etched mirrors, were compared. The key feature of our work is the use of the same laser bar during the consecutive stages of the etching experiment. The laser bar was not mounted and was studied by a bar tester under pulsed operation conditions. Between the etching stages, the scanning electron microscope (SEM) images of the same facet were registered. The measured device parameters prove that within the proposed approach, it is possible to obtain comparable optoelectronic properties of nitride-based etched and cleaved laser diodes. This is the cornerstone for the fabrication of photonic integrated circuits for visible light communication [24], especially for underwater communication [25], using wavelength division multiplexing [26]. Light sources with true, single-mode emission use evanescent mode coupling to an external ring waveguide cavity [27], and there are also light sources for on-chip bio-lab for optogenetics applications [28].

## 2. Laser Diode Preparation

The fully processed laser epi-wafer served as a starting point of the described fabrication process. The design of the laser diode used in this study is shown in Figure 1. The ridge has a width of 2 µm at the base of the mesa structure. The length of all presented devices is the same at 750 µm. More details on the epi-growth of the laser diodes used in this study can be found in the Reference [20]. Part of the epi-wafer was cleaved off to provide reference devices. Additional operations required for etched-facet fabrication are shown in Figure 2. The process starts as the formation of the hard mask consisting of a thin (0.5 μm) layer of photoresist (nLOF 2005) and an over layer of SiO_2_ with a thickness of 700 nm. The mask was completed with a negative resist (15nXT) with a thickness of about 6 μm. The mask had the form of a multi-striped pattern, of which the edges were defined at the localization of the future laser facets. After the exposure and development of the top resist, dry etched was performed on the structure in SF_6_ plasma to remove the SiO_2_ layer and then in O_2_ plasma to remove the thin photoresist at the bottom. Finally dry etching of the semiconductor layers was performed, using ICP RIE with Ar/Cl_2_ plasma. The depth of the etch was set to be about 4 um deep so that the whole epitaxial structure was etched through. As a final step, the hard mask was removed by Dimethyl sulfoxide (DMSO) lift-off.

The main focus of this work was the optimization of the wet etching stage of the facet processing. For that, the same laser chip was used, which was treated by wet etching. To control this step, the wet etching process was divided into four steps, which allowed the following total times of wet etching to be reached: 0 min, 20 min, 40 min, and 60 min. After each step, the laser was placed in a scanning electron microscope (SEM) for the facet control (SEM imaging), and its opto-electric characteristics were registered. Wet etching was always performed without using additional masks, as tetramethylammonium hydroxide (TMAH) etches mostly a- and m-planes, and additionally, the c-plane is covered with oxide and undergoes metallization.

## 3. Results and Discussion

### 3.1. Reactive Ion Etching

Before starting the main process of facet fabrication, the calibration of the dry etching procedure was conducted. Reactive ion etching is the most common method of etching c-plane-oriented GaN-based layers. The example of our optimized ICP-RIE procedure is shown in Figure 3a. The etching was done on a laser epitaxial structure but without processing (no contacts or insulation layers).

Figure 3a shows that the smoothness of the etched facet is good, but the inclination angle is about 86°, which is not sufficient to form an effective laser cavity. The estimation of the inclination of the “vertical” facet was measured for multiple samples. The tools used for this procedure were a stylus profilometer, SEM, and simple trigonometric calculations. The method was as follows. Firstly, the value of the height of the deep etch was measured by a stylus profilometer. Next, the specimen was cleaved along the easy-cleavage m-plane and placed in the SEM chamber. The sample was positioned so that the value of the height of the deep etch reproduced the result from the profilometer. This ensured the correctness of the SEM imaging and allowed us to determine the angle of the “vertical” facet inclination.

Three different etching procedures were performed for this set of parameters (85 °C, 20 min) to check if the results were repeatable. For each etching procedure, there were about 20 measurements of angles. The average is the value presented in this work. The deviations are within the estimated accuracy of this measuring method (about 1°). The obtained value of 86° is in agreement with previous literature reports [22].

The inclination angle of the RIE-etched facet is dependent on many factors, including the very details of the mask used, which makes it difficult to obtain perfect verticality. To solve this challenge, there is a need for additional anisotropic etching to obtain good verticality of the facet. This is obtained through the TMAH wet etching procedure, Figure 3b, introduced in the following section. It should be noted that Figure 3a,b shows the same sample to demonstrate problems with RIE-etched facets and the effects of TMAH etching.

### 3.2. TMAH Etching

In the developed wet etching procedure, 25% wt. aqueous TMAH solution was used. The solution was kept at a constant temperature of 80 °C and stirred during all process steps. As shown before [17,29,30], under similar conditions, etching is very fast for a-planes, very slow for m-planes, and does not occur (in a measurable way) for the c-plane. Since the laser diodes are oriented in the same direction as cleaved devices, the slowly etched m-planes will create device facets. Strict crystallographic orientation of these facets provides our devices with the high-quality resonance cavity needed for Fabry–Perot edge emitting lasers.

Figure 4a–d shows the SEM images of the etched facets with gradually increased TMAH etching time. The first stage, which is only RIE etching (Figure 4a), is characterized by relatively rough and non-vertical facets. Regarding the difference between the etch results shown in Figure 3a and Figure 4a, in Figure 3a, a much smoother facet can be seen. In the case of Figure 4a, the etching was performed on a fully processed structure, consisting of not only semiconductor layers but also metal and oxide layers. The presence of the metal and oxide layers leads to micromasking negatively influencing the smoothness of the facet. However, the increasing TMAH etching time makes the m-oriented facet smooth and vertical (Figure 4b). There is a small improvement between the 20 and 40 min TMAH etch. At times longer than 40 min, the laser ridge is gradually degraded by the formation of new m-plane regions visible at the edges of the ridge (Figure 4d). This is a negative effect as it reduces the size of the flat front facet and may lead to optical mode hopping during the operation of the laser. Clear damage done to the metallization, and the isolation of the side walls of the laser ridge can also be observed in Figure 4d.

While working on optimizing the final etched lasers, it became clear that to improve the manufacturing quality of etched facet lasers, it would be beneficial to change the workflow. The optimal workflow should include moving the metal deposition step to after the wet etching procedure.

### 3.3. Electro-Optical Testing of Etched Devices

To understand the challenge of the electro-optical characterization of prepared laser diodes, let us briefly discuss their geometry. The etch of mirrors was performed on the whole laser diode wafer. As the laser bar separation cannot be performed exactly along the etched facet formation line, there is a sizeable distance between the etched facet and physical termination of the laser bar or laser chip. The geometry of the final device is shown in Figure 5a. The distance *H* between the etched plane and the center of the active area of the laser is around 2.5 μm, while distance *L* is around 1000 μm. Simple calculation of the light escape angle, $\alpha =\tan^{-1}\frac{L}{H}\approx \tan^{-1}\frac{1000\;\mu m}{2.5\;\mu m}\approx 89.86{}^{\circ}\approx 90{}^{\circ},\;$ indicates that the angle is very close to 90°, meaning that only 50% of emitted light can be coupled to the external detector.

For optoelectronic characterization of single bars of lasers, a custom-made tester was used (Figure 5a) that consists of a movable stage with precise position control—I, microscope for visual inspection, needle manipulator for p-side contact—II, lens with fiber to collect light—III, and integrating sphere with a power meter optically coupled to the optical fiber—IV. The device power supply (V) is controlled by the computer VI, which also gathers data from the integrating sphere. All tests were conducted under the same environmental conditions (air-conditioned room with a stable temperature of 21 °C). For ease of measurement, all devices were tested using the pulsed operation mode of the power supply.

Figure 6 shows that even the laser with only the RIE-etched facet can be driven to lasing, although at high current and with a minimal slope efficiency. Even short-duration (20 min) TMAH etching improves the threshold current and efficiency, but at 40 min, the parameters of the cleaved laser become very close (within 10%) to those of the cleaved reference. Further TMAH etching does not lead to parameter improvement but introduces modal instability to laser emission—a kink-like structure in the L-I curve, which is probably related to waveguide or electric contact damage, which is also visible in SEM images, please compare Figure 4c,d.

Table 1 summarizes the described results. As seen in these results (SEM pictures and threshold currents), this is clearly an optimization problem. The parameters of this problem are the doping of the epitaxial structure (the diode structure etches slower than pure *n*-type), temperature, concentration of TMAH, and the etch time. The results shown in this publication were optimized for specific parameters. Therefore, they are probably not the best for every case where vertical facets are desirable. However, this set of parameters can be used as a starting point for the optimization of such a process for similar epitaxial structures.

## 4. Conclusions

This paper presents the fabrication of (Al, In) GaN laser diodes with etched facets grown on bulk GaN substrates. The facets were obtained by a two-step etching process: dry etching by reactive ion etching followed by wet etching in TMAH. The work focused on optimizing the duration of the process and its effect on facet morphology and optoelectronic parameters.

The key element of this study was examining the evolution of the parameters of the same laser diode chips undergoing consecutive steps of wet etching. During this experiment we:Demonstrated the optimal parameters of wet etching smoothing the dry-etched facet (40 min in a 25% wt. aqueous solution of TMAH at 80 °C).Showed the influence of the elements of a fully processed laser structure (in a classical way) on the morphology of the etched facet.Concluded that the order of the processing steps of the lasers needs to be adjusted when using wet etching to limit the negative influence on the bottom contact (N-side) and wet-etched facet shape.Demonstrated the comparison of optoelectrical parameters of the laser diodes after consecutive etching steps.Compared the results with the cleaved counterparts.

This study proves that it is possible to obtain lasers with etched facets that have comparable threshold currents to their cleaved counterparts.

## Figures and Tables

**Figure 1 micromachines-14-00408-f001:**
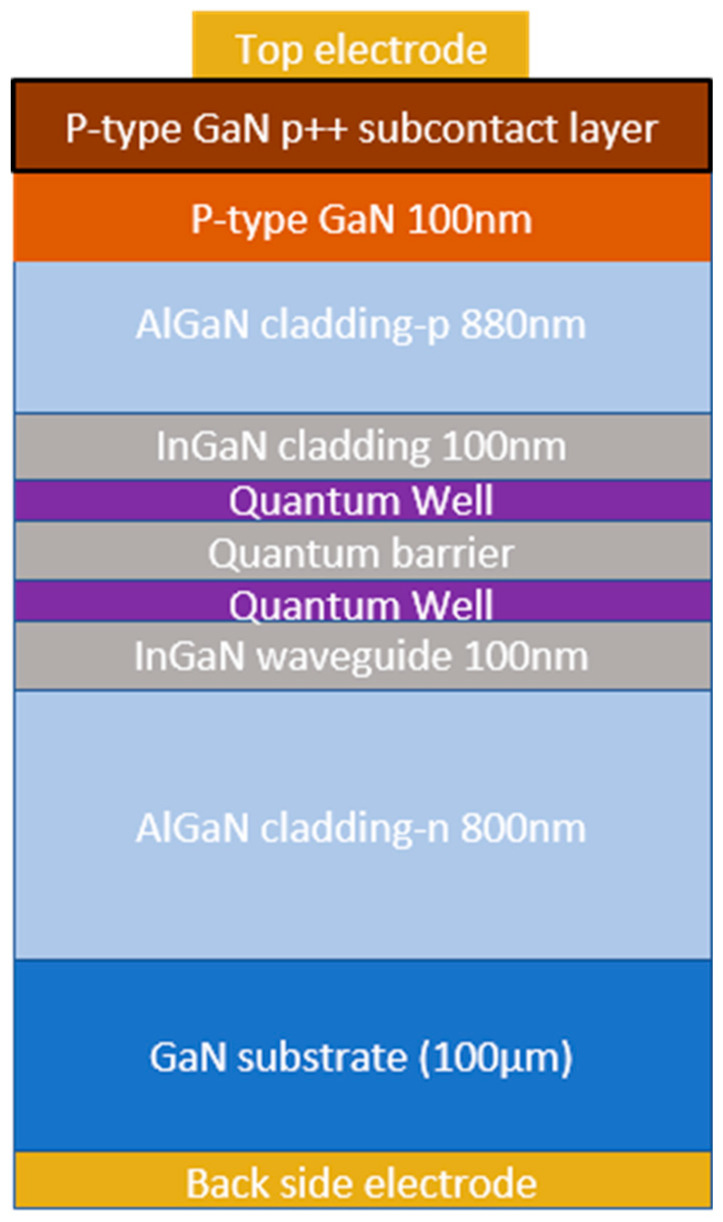
Schematic cross-section of the epitaxial structure of laser diodes used in the present study [20].

**Figure 2 micromachines-14-00408-f002:**
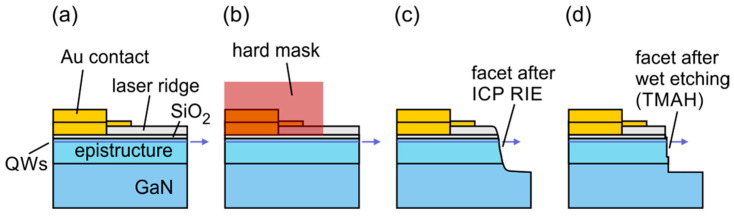
Schematic representation of the processing steps: (**a**) Epitaxial structure processed in the conventional way [20], (**b**) preparation of the hard mask used to deep etch the laser facet using RIE, (**c**) cross-section after the dry etching procedure with an exaggerated sloped facet, (**d**) device after the optimal wet etch step with a fully formed facet.

**Figure 3 micromachines-14-00408-f003:**
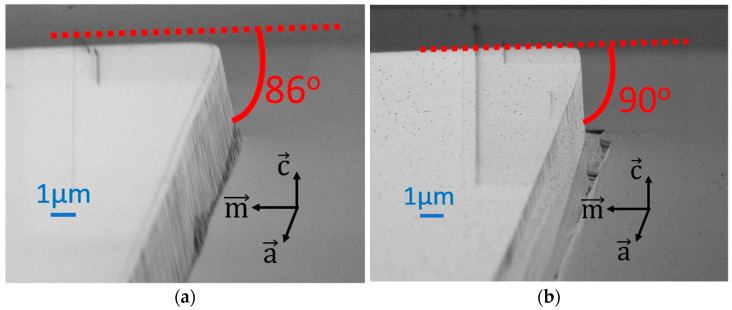
SEM images of the test sample of the laser epitaxial structure (**a**) after reactive ion etching, (**b**) the same edge after RIE and TMAH wet etching.

**Figure 4 micromachines-14-00408-f004:**
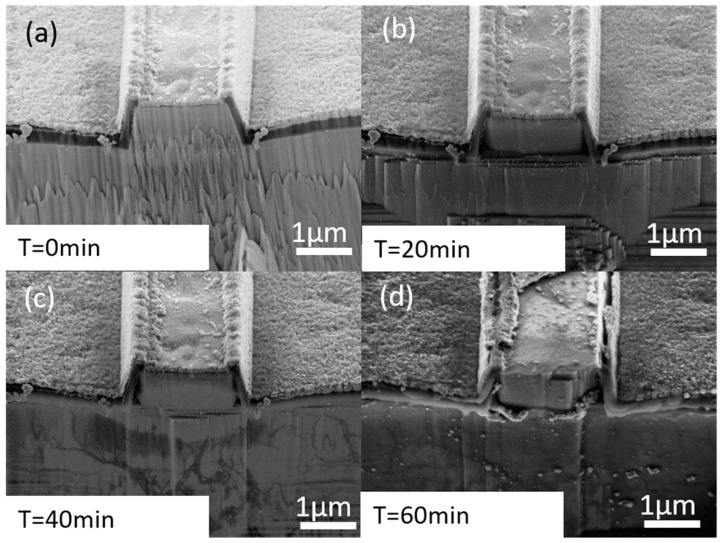
SEM images of the laser facet after: (**a**) only RIE, (**b**) RIE + 20 min TMAH, (**c**) RIE + 40 min TMAH, (**d**) 60 min TMAH.

**Figure 5 micromachines-14-00408-f005:**
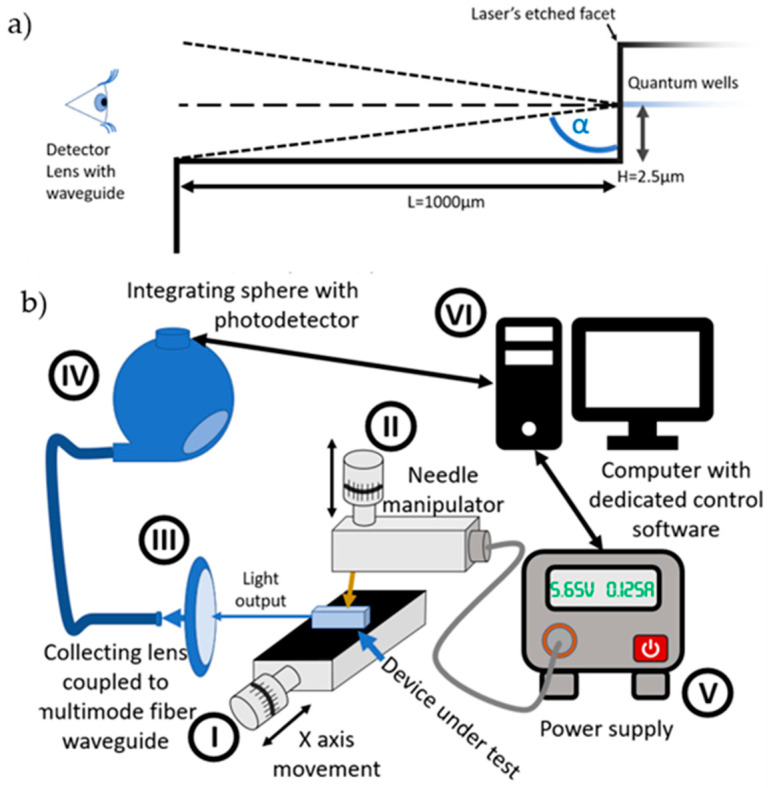
Schematic representations of: (**a**) The geometry of the sample and its effects on reducing light output from a single etched facet chip and (**b**) The opto-electrical testing setup.

**Figure 6 micromachines-14-00408-f006:**
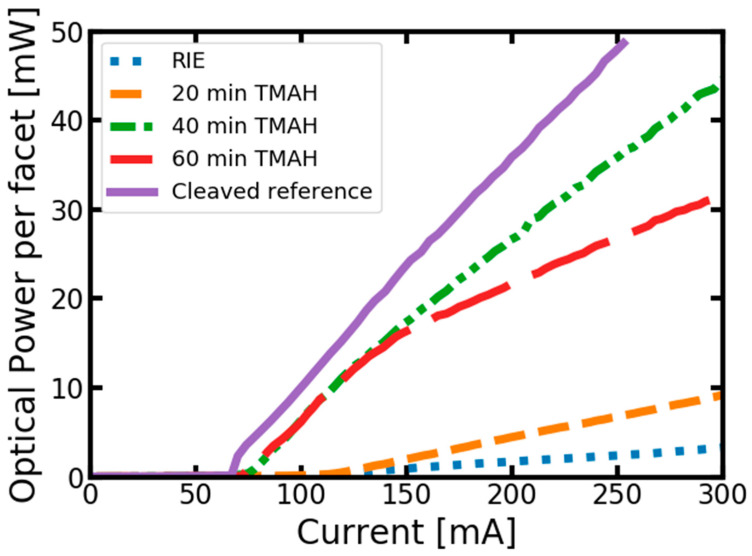
Light current characteristic of differently processed laser diodes (pulse current measurements).

**Table 1 micromachines-14-00408-t001:** Lasing parameters of etched facet laser diodes.

Etching Procedure	Threshold Current (mA)	Slope Efficiency (W/A)
Cleaved reference	67	0.242
RIE	112	0.015
RIE + 20min TMAH@80 °C	106	0.048
RIE + 40min TMAH@80 °C	75	0.181
RIE + 60min TMAH@80 °C	72	0.102

## Data Availability

Data are available upon contacting the corresponding author.

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
