# Peer review of "InGaN Laser Diodes with Etched Facets for Photonic Integrated Circuit Applications"

_micromachines, 2023, doi:10.3390/mi14020408_

Round 1

Reviewer 1 Report

The manuscript entitled ”  InGaN laser diodes with etched facets for photonic integrated circuit applications “  by  Krzysztof Gibasiewicz et al. reports optimizing the processing schemes for etched facet lasers and studying their work parameters. The manuscript is a good piece for one who wants to know the critical enabling step for the future fabrication of photonic integrated circuits with on-chip integrated laser diodes and ultra-short cavity lasers. The authors have performed a fair investigation on this paper. The language used in this article is good. Overall, the report is decent. However, all the statements need a small amount of clarification by providing detailed information to improve the fineness of this manuscript. I want to address general queries on this review manuscript, which will help improve the quality of the article. Please find the comment below.

  1. Page no 1. Line no 11, in the abstract, please provide the abbreviation of Reactive ion etching (RIE) and Tetramethylammonium hydroxide (TMAH)
  2. Page no 1, line no 28 “ The main motivation to form the mirrors by etching 28 was and still is the fabrication of short cavity lasers (short lasers are difficult to cleave) 29 and photonic integrated circuits with fully integrated light sources [*]. Please provide a suitable reference if you have any.
  3. Page no 1, line no 31, There are two fundamental problems related to etched laser mirrors technology, namely verticality of the mirrors and their smoothness[*]. Please provide a suitable reference if you have any.
  4. Page no 1, line no 33, Verticality requires the etch process to be sensitive to the crystallographic orientation of the facet, smoothness may be problematic in case of etching multilayer structure (varying chemical composition) of the laser diodes [*]. Please provide a suitable reference if you have any.
  5. Page no 3, line no 78, “Wet etching was always performed without using additional masks, as Tetramethylammonium hydroxide (TMAH) etches mostly a- and m-planes, and additionally, c-plane is covered with oxide and metallization” [*]. Please provide a suitable reference if you have any.
  6. Page no 3, line no 83, Reactive ion etching is the most common method of etching c-plane oriented GaN-based layers. [*]. Please provide a suitable reference if you have any.
  7. Page no 3, line no 95, How do authors estimate the angle 860 in figure 3 a? 
  8. Page no 4, line 123, “One can ask about the difference between the etch results shown in Fig. 3a and Fig 4a – in Fig 3. we see a much smoother facet”, What does the author mean by facet here? Which plane?
  9. In figure 5, what does the author mean by detector lens with waveguide? Also, please provide a suitable reference if you have any.
  10. In figure 5, the sides H and L include a right-angled triangle, meaning one of the angles is 900 if α~90; how does it satisfy the trigonometry condition? Why did the author choose L=1000 µm
  11. Page no, 5 line 148, It consists of a movable stage with precise position control, a microscope for visual inspection, a needle manipulator for p-side contact, a lens with fiber to collect light and optical power meter [*]Please provide a suitable reference if you have any?
  12. Figure 6 shows cleaved reference has better slope efficiency than RIE + xmin TMAH@80oC. The manuscript reports optimising an etched facet by dry etching with RIE and wet etching with TMAH. Does the author's motivation meet the investigative result? 
  13. The first two references [1,2] were pretty old, and it would be good to provide recent research work related to the statement. 

Author Response

Thank You for Your patience. Please find our response to Your review in the attached Word file.

Best regards

Reviewer 2 Report

Review for Micromachines 2022 (micromachines-2138977-peer-review-v1):

The authors presented a manuscript with the title of “InGaN laser diodes with etched facets for photonic integrated 2 circuit applications”. The authors demonstrated the fabrication of (Al,In)GaN laser diodes with etched facets. This work may be interested to the researchers/readers in the related community. 

However, the Reviewer cannot recommend the publication of this manuscript (at least this version). The major issues of this manuscript are listed as below:

1. The title: “InGaN laser diodes with etched facets for photonic integrated circuit applications”; which application InGaN laser can be used in photonic integrated circuit?

2. The basic laser information was not presented in this manuscript; for example, what is the laser ridge width? What is the laser cavity length? What is the testing temperature? The laser was tested under continuous wave (CW) or pulsed condition?

3. Page 2: line 70, “As a final step, hard mask was removed by DMSO lift-off of.” What is DMSO? The sentence needs to be re-arranged.

4. Fig. 1: the schematic may not be completed. For example, there is no contact layer in the laser structure.

5. Fig.2: the process description is not clear;

6. Table 1: the slope efficiency is from one facet or both facets? And how many devices per condition has been tested? Are they repeatable?

Due to above issues, The Reviewer feels reluctant to recommend publishing this paper (at this version without detailed info). 

Author Response

Thank You for Your patience. Please find our response to Your review in the Word file below. Best regards.

Reviewer 3 Report

Please, see the file attached.

Author Response

(The authors gave the same response as above.)

Round 2

Reviewer 1 Report

The authors have submitted a revised manuscript,” InGaN laser diodes with etched facets for photonic integrated circuit applications  “ by  Krzysztof Gibasiewicz et al. reports the optimization methods of fabricating (Al, In)GaN laser diodes with etched facets. Authors have significantly improved in illustrating their research work in terms of figures and contents. Also, the authors are given an almost satisfactory response to the comment raised. Now the article is in acceptable form for publication.  

Author Response

Thank You for your time to review our manuscript. We believe that because of your valuable input, our manuscript has improved significantly.

Best regards.

Reviewer 2 Report

All my comments for the previous version have been fully addressed/corrected. I don’t have further comments/questions. I believe this draft has been revised carefully and I therefore recommend the publication of this draft.

Author Response

(The authors gave the same response as above.)

Reviewer 3 Report

Please, see the file attached.

Author Response

Thank You for your time to review our revised manuscript. Please find our response in the Word files below.

Best regards
